# Anxiety among cancer patients: A cross-sectional study at The University of Gondar comprehensive specialized hospital

**Hailu Aragie**[1]*, **Dagnew Getnet Adugna**[1], **Nega Dagnew Baye**[1], **Habtu Kifle Negash**[1,2]

**1** Department of Human Anatomy, School of Medicine, College of Medicine and Health Science, University of Gondar, Gondar, Ethiopia, **2** Department of Epidemiology and Biostatistics, Institute of Public Health, College of Medicine and Health Science, University of Gondar, Gondar, Ethiopia

* hailuaragie995@gmail.com

## Abstract

### Introduction

Anxiety is a prevalent psychological issue among cancer patients, significantly affecting their quality of life and potentially influencing treatment outcomes. This study aims to investigate the prevalence and associated factors of anxiety among cancer patients at the University of Gondar Comprehensive Specialized Hospital in Northwest Ethiopia.

### Methods

A cross-sectional study design was used, involving 384 cancer patients, selected by systematic random sampling technique, from the oncology ward. Data were collected using the Generalized Anxiety Disorder 7-item (GAD-7) scale and the Oslo Social Support Scale (OSS-3). Descriptive and inferential statistics, including bi-variable and multi-variable logistic regression analyses, were employed to analyze the data.

### Results

The prevalence of anxiety was 88.3%, with 34.4% of patients experiencing severe anxiety, 29.2% moderate anxiety, and 24.7% mild anxiety. Significant predictors of anxiety included participants age>= 61years (AOR = 2.99, 95% CI = 1.31–6.79), participants age 30-60 years (AOR = 4.13, 95% CI = 1.53–11.07), female sex (AOR = 2.04, 95% CI = 1.02–4.07), advanced cancer stage (AOR = 3.08, 95% CI = 1.11–8.53), and ongoing treatment (AOR = 2.40, 95% CI = 1.14–5.10).

### Conclusion

The study found a high prevalence of anxiety among cancer patients, compared to the previous studies, highlighting the need for integrated psycho-oncology services. Specific interventions are necessary for high-risk groups, including older patients, females, and those with advanced-stage cancer.

**Data availability statement:** According to the ethical review committee, the data contains potentially identifying characteristics as well as sensitive patient information and cannot be shared publicly. All data requests can be sent to the Director of the School of Medicine (Dr. Mulugeta Ayalew at mulugetaayalew336@gmail.com), or the Chair, the Ethical Review Board (prof. Abebe Muche at abemuche14@gmail.com).

**Funding:** The author(s) received no specific funding for this work.

**Competing interests:** The authors declare that they do not have any conflicting of interest.

**Abbreviations:** AOR, Adjusted Odd Ratio; BMI, Body Mass Index; CI, Confidence Interval; COR, Crude Odd Ratio; CA, Cancer; DM, Diabetes Mellitus; GI, Gastrointestinal; GU: Genitourinary

## Introduction

Cancer ranks as the second leading cause of death globally, responsible for 9.6 million deaths in 2018, with over 70% of these fatalities occurring in low and middle-income nations [1]. In Ethiopia, the Federal Ministry of Health (FMOH) estimates that more than 150,000 new cancer cases are diagnosed annually, accounting for 4% of all deaths in the country and Sub-Saharan Africa [2].

The diagnosis and treatment of cancer are often accompanied by substantial psychological distress, including anxiety, which can profoundly affect patients' quality of life and treatment outcomes [3,4]. Anxiety, a prevalent condition among cancer patients, is characterized by a range of physiological and psychological symptoms. Physically, it may present as increased heart rate, rapid breathing, excessive sweating, and tremors. Psychologically, it manifests as intense feelings of apprehension, powerlessness, and fear of losing control [5,6]. These symptoms can further complicate the already challenging experience of navigating cancer treatment, leading to a reduced ability to cope with the illness.

Research has shown that about 30% of cancer patients endure some form of mental distress during their treatment journey, with approximately 19% specifically experiencing anxiety symptoms [7,8]. These findings underscore the substantial psychological burden faced by cancer patients. Anxiety can arise from various stressors, including uncertainty about treatment outcomes, the potential for recurrence, and the fear of mortality, all of which can compound the emotional toll of the disease.

Although the physical challenges of cancer and its treatment are widely recognized, the psychological impact—particularly anxiety—remains less explored, especially in low-resource settings like Ethiopia. In such environments, the focus on the psychological well-being of patients may be overshadowed by the immediate demands of managing physical symptoms and ensuring access to treatment. The limited availability of mental health services, coupled with cultural stigmas surrounding mental illness, can further exacerbate the problem.

Understanding the prevalence and contributing factors of anxiety among cancer patients is critical for developing effective, tailored interventions that address this aspect of patient care [6,9,10]. In low-resource settings, such insights can inform strategies to integrate psychological support into cancer care, ensuring that healthcare providers consider both the physical and emotional dimensions of the disease. By doing so, healthcare systems can enhance the overall quality of care and improve cancer patients' treatment experience and outcomes.

## Methods and materials

Ethical clearance was obtained from the University of Gondar School of Medicine Review Board. written informed consent was secured from all participants, ensuring confidentiality and anonymity.

### Study design and setting

This study employed a cross-sectional design and was conducted at the University of Gondar Comprehensive Specialized Hospital, a major tertiary care center in Northwest Ethiopia. The hospital serves a large catchment area and provides comprehensive oncology services, including diagnosis, treatment, and follow-up care.

### Study population, sample size, and sampling technique

The study population comprised all cancer patients attending the oncology ward at the University of Gondar Comprehensive Specialized Hospital between January 10 and March

10, 2024. The sample size for the study was determined using a single population proportion formula, considering a 95% confidence level, a 5% margin of error, and a previously reported prevalence of anxiety (51%) [11]. Accordingly, the final sample size with a 10% non-response rate was 403 participants, who were selected using a systematic random sampling method with a skip interval of 1.

### Inclusion criteria

All adults diagnosed with any types of cancer and receive treatment or follow-up care in this hospital.

### Exclusion criteria

Patients with severe cognitive impairments or psychiatric disorders that could interfere with their ability to participate in the study.

### Variables in the study

**Dependent variable.** ✓        Anxiety

**Independent variables.**  Socio-demographic: Age, Sex, Residence, Occupation, income clinical: Type of cancer, stage of cancer, site of cancer, duration from diagnosis, Presence of comorbidity (DM, HTN), type of cancer treatment, type and number of analgesics, presence of metastasis, duration of medication use, length of stay before starting treatment Psycho-social and behavioral factors: Social support, history of CMD, alcohol consumption, cigarette smoking, physical activity.

### Operational definitions

**Anxiety.**  Anxiety was measured using the Generalized Anxiety Disorder 7-item (GAD-7) scale [12]. The GAD-7 questionnaire is a brief and widely used tool for measuring generalized anxiety disorder. It assesses the frequency of symptoms that the respondent has been bothered by over the past two weeks. The questionnaire consists of items rated on a scale from 0 (not at all) to 3 (nearly every day). To determine a cut-off point that balances sensitivity and specificity, a receiver operating characteristic (ROC) curve analysis was performed. This analysis helps identify the optimal threshold score on the GAD-7 that maximizes the tool's ability to accurately detect anxiety in the study population.

**Social support.**  Social support was measured by the Oslo Social Support Scale (OSSS-3). OSSS-3 has three items measured by Likert scales, which are summed to 14 points and categorized as 'poor' if the total score is 3–8, moderate 9–11, and strong 12–14 [13].

### Data collection procedures

Data were collected using pretested and structured interviewer-administered questionnaires and from chart review. The questionnaire consists of three parts: The first is for socio-demographic and behavioral characteristics including age, gender, educational attainment, residence, marital status, income level, occupational status, smoking, alcohol, and physical activity of the study participants. The second part is for, questions about the clinical and medical history of a patient including the stage of the disease, type of current treatment, time of diagnosis, and type of cancer. The third part focused on the questionnaire to assess social support and the final part of the questionnaire contain questions to assess anxiety. Information on the variables like type and stage of cancer, type of treatment, and type of analgesics

were collected from patient charts. The data were collected by General Practitioners who were working in the oncology ward and were supervised by the principal investigator.

### Data quality assurance

The quality of data was ensured through training of data collectors, close supervision, and prompt feedback. The training, was given for 01 day, consisted of instruction on Interview techniques as per the prepared tool. The data were checked for any inconsistencies, coding errors, out-of-range, completeness, accuracy, clarity, and missing values, and appropriate corrections were made by the principal investigator consistently on a daily basis.

### Data processing and analysis

The survey data were entered into EPI-INFO version 7 and analyzed by STATA 14 software. Descriptive statistics were used and presented using texts, graphs, and tables. A logistic regression model was used to identify factors affecting anxiety. Both bi-variable and multivariable logistic regression models were carried out. Variables with a p-value of less than 0.2 in the bi-variable analysis were entered into the multivariable analysis. Both Crude Odds Ratio (COR) and Adjusted Odds Ratio (AOR) with 95% confidence intervals were estimated to show the strength of associations. Finally, a p-value of less than 0.05 in the multi-variable logistic regression analysis was used. For this study, the Hosmer and Lemeshow goodness-of-fit test was applied, yielding a value of 0.414. Multicollinearity was also assessed using the Variance Inflation Factor (VIF) and tolerance through a pseudo-linear regression analysis. All independent variables had VIF values less than 5 and tolerance values greater than 0.1, indicating the absence of multicollinearity.

### Ethical considerations

Ethical clearance was obtained from the University of Gondar School of Medicine Review Board. written informed consent was secured from all participants, ensuring confidentiality and anonymity.

## Result

### Socio-demographic characteristics of the study participants

A total of 384 cancer patients participated in the study, yielding a response rate of 95.3%. The ages of the respondents ranged from 18 to 82 years, with a mean age of 49.3 years and a standard deviation of 13.0 years. The majority of patients, 64.4%, were from rural areas. More than half of the respondents were married, and Orthodox Christianity was the predominant religion, with 63.3% of participants identifying as such. Over one-third of the respondents were farmers, and nearly two-thirds fell into the middle-income category (Table 1).

### Cancer type and related characteristics

Over two-thirds of the patients in the study were diagnosed with cervical or gastrointestinal (GI) cancer. Stage four and stage three cancers were present in more than 12% and 15% of the participants, respectively. Three-fifths of the cancer patients had been diagnosed within the past four weeks. The majority of patients, 82.5%, had not yet started cancer treatment at the time of the study. Additionally, more than 30% of the participants had metastasized cancer. About 60% of the participants reported experiencing cancer pain, but only 41.2% were using pain management medication. Approximately half of the respondents reported having moderate social support (Table 2).

**Table 1. Descriptive statistics of socio-demographic characteristics of adult cancer patients evaluated at the oncology unit, in the University of Gondar Comprehensive Specialized Hospital, northwest Ethiopia, 2024 (n = 384).**

| Variables | Categories | Frequencies | Percent |
|---|---|---|---|
| Age | 18–30 | 99 | 25.8 |
| | 31-60 | 215 | 56 |
| | ≥ 61 years old | 70 | 18.2 |
| Residence | Urban | 136 | 35.6 |
| | Rural | 246 | 64.4 |
| Marital status | Married | 191 | 49.7 |
| | Unmarried | 193 | 50.3 |
| Religion | Orthodox | 243 | 63.3 |
| | Muslim | 91 | 23.7 |
| | Protestant | 45 | 11.7 |
| | Others | 5 | 1.3 |
| Occupation | Farmer | 144 | 37.5 |
| | Merchant | 82 | 21.3 |
| | Government worker | 119 | 31 |
| | Others | 39 | 20.2 |
| Income | Low | 112 | 29.2 |
| | Medium | 248 | 64.5 |
| | High | 24 | 6.3 |

## Level of anxiety and comorbidity

Among our study participants, 88.3% (85.4–91.8) experienced anxiety, with 34.4% (20.1–41.9) reporting severe anxiety, 29.2% (24.3–33.3) moderate anxiety, and 24.7% (14.8–29.7) mild anxiety, as indicated in Table 3. Additionally, about one-third of the cancer patients were hypertensive, and 16.7% were diabetic. Nearly 20% of the participants were either under-weight or overweight, according to our BMI assessment. The study also found that 52.3% of the participants were alcohol users, and 16.2% were smokers.

## Factors associated with anxiety among cancer patients

In the binary logistic regression analysis, several variables were identified as candidates for multi-variable analysis based on a p-value of less than 0.2 for anxiety. These variables included the age of cancer patients, marital status, sex, cancer treatment, stage of cancer, cancer pain, and anti-pain use (Table 4).

In the subsequent multi-variable logistic regression analysis, age, sex, cancer treatment, and stage of cancer were found to be significantly associated with anxiety. Cancer patients aged 61 years and older had approximately three times higher odds of experiencing anxiety compared to those aged 18–30 years (AOR = 2.99, 95% CI = 1.31–6.79). Similarly, patients aged 31–60 years had more than four times higher odds of having anxiety compared to those in the 18–30 age group (AOR = 4.13, 95% CI = 1.53–11.07). Patients who had started cancer treatment had 2.4 times higher odds of developing anxiety compared to those who had not yet undergone treatment (AOR = 2.40, 95% CI = 1.14–5.10). Participants with stage 3 or 4 cancers had over three times higher odds of developing anxiety than those with stage 1 cancer (AOR = 3.08, 95% CI = 1.11–8.53). Female cancer patients had about twice the odds of experiencing anxiety compared to male patients (AOR = 2.04, 95% CI = 1.02–4.07).

**Table 2.** Descriptive statistics of cancer type and related characteristicsamong adult cancer patients at oncology unit, in the University of Gondar comprehensive specialized hospital, northwest Ethiopia, 2024 (n = 384).

| Variables | Categories | Frequencies | Percentages |
|---|---|---|---|
| Cancer type | Cervical CA | 69 | 17.9 |
| | GI CA | 68 | 17.7 |
| | Haematologic CA | 64 | 16.7 |
| | Breast CA | 55 | 14.3 |
| | Endocrine CA | 36 | 9.4 |
| | GU CA | 25 | 6.5 |
| | Lung CA | 16 | 4.2 |
| | Skin CA | 15 | 3.9 |
| | Other types | 36 | 9.4 |
| Stage CA | Stage 1 | 221 | 57.5 |
| | Stage 2 | 56 | 14.6 |
| | Stage 3 | 59 | 15.4 |
| | Stage 4 | 48 | 12.5 |
| Duration after cancer diagnosis | <= 4 months | 233 | 60.7 |
| | > 4 months | 151 | 39.3 |
| Metastasis | No | 264 | 68.7 |
| | Yes | 120 | 31.3 |
| Cancer treatment | No | 67 | 82.5 |
| | Yes | 317 | 17.5 |
| The time interval between diagnosis and first medication | 0-3 weeks | 289 | 90.9 |
| | >= 4 weeks | 28 | 9.1 |
| Treatment modalities | Chemotherapy | 80 | 20.8 |
| | Surgery | 123 | 32.0 |
| | Chemo and surgery | 114 | 29.7 |
| | No treatment | 66 | 17.7 |
| Cancer pain | No | 154 | 40.1 |
| | Yes | 230 | 59.9 |
| Antipain use | No | 158 | 41.2 |
| | Yes | 226 | 58.8 |
| Social support | Poor | 139 | 36.2 |
| | Moderat | 188 | 49.0 |
| | Strong | 57 | 14.8 |

GI = Gastrointestinal, CA = Cancer, GU = Genitourinary.

## Discussion

The present study assessed the magnitude and associated factors of anxiety among 384 cancer patients at the University of Gondar Comprehensive Specialized Hospital (UOGCSH), Northwest Ethiopia. The findings revealed that 24.7%, 29.2%, and 34.4% of the patients experienced mild, moderate, and severe anxiety, respectively. Thus, the overall prevalence of anxiety in this study was 88.3%. This figure is notably higher than that reported in a study conducted in the same setting seven years ago [11]. The observed disparity could be due to differences in the study population, sampling methodology, and sample size. I.e. the prior study was conducted among inpatients with a small sample size and nonrandom sampling procedures. Anxiety prevalence in our study was also higher compared to previous research which was conducted

**Table 3. Descriptive statistics of the level of anxiety and comorbidities among cancer patients evaluated at the oncology unit, in the University of Gondar Comprehensive Specialized Hospital, northwest Ethiopia, 2024 (n = 384).**

| Variables | Categories | Frequencies | Percent |
|---|---|---|---|
| Level of anxiety | No anxiety | 45 | 11.7 |
| | Mild anxiety | 95 | 24.7 |
| | Moderate anxiety | 112 | 29.2 |
| | Severe anxiety | 132 | 34.4 |
| Hypertension | No | 245 | 63.8 |
| | Yes | 139 | 36.2 |
| DM | No | 320 | 83.3 |
| | Yes | 64 | 16.7 |
| BMI | Underweight | 47 | 12.3 |
| | Normal | 310 | 80.7 |
| | Overweight | 27 | 7.0 |
| Medication for comorbidity | No | 357 | 93.0 |
| | Yes | 27 | 7.0 |
| Alcohol intake | No | 183 | 47.7 |
| | Yes | 201 | 52.3 |
| Smoking | No | 322 | 83.8 |
| | Yes | 62 | 16.2 |

BMI = Body Mass Index, DM = Diabetes mellitus

**Table 4. Result of Bi-variable and multi-variable logistic regression for factors associated with anxiety among cancer patients in the University of Gondar comprehensive specialized hospital, northwest Ethiopia, 2024 (n = 384).**

| Variables | Category | Anxiety | | COR 95% CI | AOR 95% CI |
|---|---|---|---|---|---|
| | | No | Yes | | |
| Age of cancer patients | 18–30 | 8 | 91 | 1 | 1 |
| | 31–60 | 22 | 193 | 3.10 (1.24–7.79)** | 4.13(1.53–11.07)** |
| | >=61 | 15 | 55 | 2.39 (1.16–4.92)** | 2.99 (1.31–6.79)** |
| Marital status | Not married | 27 | 166 | 1 | 1 |
| | Married | 18 | 173 | 0.45 (0.19–1.08) | 1.37 (0.68–2.74) |
| Sex | Male | 29 | 158 | 1 | 1 |
| | Female | 16 | 181 | 2.08 (1.09–3.96)* | 2.04 (1.02–4.07) * |
| Cancer treatment | No | 14 | 31 | 1 | 1 |
| | Yes | 53 | 286 | 2.44 (1.22–4.89)** | 2.40 (1.14–5.10) * |
| Stage of cancer | Stage 1 | 36 | 185 | 1 | 1 |
| | Stage 2 | 3 | 53 | 3.44 (1.02–11.60)* | 3.19 (0.90–11.37) |
| | Stage 3 and 4 | 6 | 101 | 3.28 (1.33–8.03)** | 3.08 (1.11–8.53) * |
| Cancer pain | No | 24 | 130 | 1 | 1 |
| | Yes | 21 | 209 | 1.84 (0.98–3.43) | 2.02 (0.88–46.35) |
| Anti pain use | Yes | 21 | 205 | 1 | 1 |
| | No | 24 | 134 | 0.57 (0.31–1.07) | 1.11 (0.05–25.39) |

AOR = Adjusted Odd Ratio; CI = Confidence Interval, COR = Crude Odd Ratio.

***p-value =< 0.001, **p value =< 0.01,*p-value =< 0.05.

Hosmer and Lemeshow test goodness of fit = 0.414.

elsewhere [9,10,14,15]. This elevated prevalence may be linked to the various physical and psychological impacts of cancer and its treatments, such as fatigue, weight loss, alopecia, and surgical complications like colostomy and mastectomy [16].

Our findings underscore the impact of the cancer stage on anxiety levels. Consistent with previous research [10,17], advanced cancer stages were associated with increased anxiety. Patients with advanced cancer often face greater physical debilitation, symptoms, and metastases, alongside difficulties with eating, drinking, pain, and sleep deprivation. These conditions contribute to heightened anxiety due to concerns about health, diminished hope for effective treatment, frequent thoughts of death, and worries about family members. Additionally, the surgical treatment for advanced cancer involves pain and prolonged recovery, further exacerbating anxiety among patients.

Anxiety was shown to be more common in middle-aged (30–60 years) and elderly (>= 61 years) people compared to younger ages in our study. According to a previous study, people's psychological reactions to cancer differ depending on their age I.e. when a person gets older his/her fear of cancer disease increases, and this, in turn, leads to anxiety. On the other way, older patients experience longer disease duration, a higher risk of cancer metastases, and more disability, all of which contribute to increased anxiety.

Analysis of anxiety scores in our study revealed that females have higher anxiety ratings compared to males. This finding is supported by several other studies [18-20]. However, research on the relationship between sex and anxiety has yielded varied results [21]. Experimental studies suggest that the increased mood changes and anxiety observed in female patients may be linked to fluctuations in estrogen and progesterone levels, which can influence emotional regulation and stress responses [22].

In the present study, patients' status of cancer treatment was found to be associated with anxiety. Patients who started cancer treatments were more affected by anxiety than patients who did not start cancer treatments. The effect of various treatments on the look of the patient Due to malaise, weight loss, baldness, and surgical consequences such as colostomy and mastectomy, these people experience a lot of anxiety[16].In some tumors, targeting single molecular aberrations or cancer pathways has resulted in good clinical responses that have had a minor impact on survival. However, this reductionist approach to cancer treatment remains, and numerous hurdles must be overcome to enhance treatment outcomes for patients[11].

## Limitations

There are certain limitations to the current study. Because we only used one research site, the findings cannot be applied to all cancer patients in Ethiopia. Patients getting curative and palliative care were included in the study, but their emotional states are likely to differ greatly.

## Conclusion

The high prevalence of anxiety among cancer patients at the University of Gondar Comprehensive Specialized Hospital underscores the need for integrating psychological support into standard cancer care. Addressing identified risk factors through targeted interventions can significantly improve patient outcomes and quality of life.

The University of Gondar Hospital should integrate psycho-oncology services within its cancer care departments to provide essential counseling and psychological support to patients. Additionally, the Ministry of Health, in collaboration with various governmental and non-governmental organizations, should offer training programs to enhance healthcare providers' skills in identifying and managing anxiety. Policymakers should also prioritize the

creation of supportive care programs that focus on improving social support systems, which can play a crucial role in reducing anxiety among cancer patients.

## Acknowledgments

We would like to thank all study participants and data collectors for their great contributions to the success of this study. We would also like to acknowledge the University of Gondar for ethical clearance.

## Author contributions

**Conceptualization:** Hailu Aragie, Dagnew Getnet Adugna, Nega Dagnew Baye.

**Data curation:** Hailu Aragie, Dagnew Getnet Adugna, Nega Dagnew Baye, Habtu Kifle Negash.

**Formal analysis:** Hailu Aragie, Nega Dagnew Baye, Habtu Kifle Negash.

**Funding acquisition:** Hailu Aragie, Dagnew Getnet Adugna, Nega Dagnew Baye, Habtu Kifle Negash.

**Investigation:** Hailu Aragie, Nega Dagnew Baye, Habtu Kifle Negash.

**Methodology:** Hailu Aragie, Nega Dagnew Baye, Habtu Kifle Negash.

**Project administration:** Hailu Aragie.

**Resources:** Hailu Aragie, Habtu Kifle Negash.

**Software:** Hailu Aragie.

**Supervision:** Hailu Aragie.

**Validation:** Hailu Aragie.

**Visualization:** Hailu Aragie.

**Writing – original draft:** Hailu Aragie.

**Writing – review & editing:** Hailu Aragie, Dagnew Getnet Adugna.

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
