## [Decision Letter · Decision Letter 0]

10 Oct 2024

PONE-D-24-31541Anxiety among Cancer Patients: A Cross-Sectional Study at the University of Gondar Comprehensive Specialized HospitalPLOS ONE

Dear Dr. Aragie,

Thank you for submitting your manuscript to PLOS ONE. After careful consideration, we feel that it has merit but does not fully meet PLOS ONE’s publication criteria as it currently stands. Therefore, we invite you to submit a revised version of the manuscript that addresses the points raised during the review process.

We look forward to receiving your revised manuscript.

Kind regards,

Zelalem Belayneh Muluneh

Academic Editor

PLOS ONE

Journal Requirements:

Reviewers' comments:

Reviewer's Responses to Questions

**Comments to the Author**

1. Is the manuscript technically sound, and do the data support the conclusions?

Reviewer #1: Partly

Reviewer #2: Yes

2. Has the statistical analysis been performed appropriately and rigorously? 

Reviewer #1: I Don't Know

Reviewer #2: Yes

3. Have the authors made all data underlying the findings in their manuscript fully available?

Reviewer #1: No

Reviewer #2: Yes

4. Is the manuscript presented in an intelligible fashion and written in standard English?

Reviewer #1: Yes

Reviewer #2: Yes

5. Review Comments to the Author

Reviewer #1: There are lots of grammatical and sentence clarity errors. Please improve grammar and spelling errors. Again, you have to include statement of declaration, conflict of interest, author contributions and funding statement. Unless it is incomplete manuscript. I will recommend carefully reading authors guidelines

Reviewer #2: Avery good study the study format is very clear,not much wordy i have justfew comments:

-Comment 1: The abrrevations i prefer to be put at the begining of the study not at the end.

--Comment 2:Reefrrences: I wish to cite more recent refrrences(2024/2023/2022).

6. PLOS authors have the option to publish the peer review history of their article (what does this mean? ). If published, this will include your full peer review and any attached files.

**Do you want your identity to be public for this peer review?** For information about this choice, including consent withdrawal, please see our Privacy Policy .

Reviewer #1: **Yes: ** Gemechis Adula

Reviewer #2: **Yes: ** Mohsen M A Abdelhafez

---

## [Author Response · Author response to Decision Letter 1]

16 Oct 2024

Date:13/10/2024

A point-by-point response to editors and reviewers

Title: Anxiety among Cancer Patients: A Cross-Sectional Study at the University of Gondar Comprehensive Specialized Hospital

Manuscript number: PONE-D-24-31541

Subject: Submitting a revised version of the manuscript

We would like to thank the reviewers and editors for sharing their views and novel scholarly experiences. The comments are very imperative and I strongly believe in improving the manuscript. The authors try to address all the comments raised by the reviewers line by line in the main document. The point-by-point responses for each of the comments, questions, and the revised manuscript are provided in the attached documents.

Thank you for considering our manuscript again

Response to the academic editor

Dear Editor, we appreciate your valuable feedback and requirements, and we have addressed them in our responses as outlined below.

Journal Requirements:

Author response: The authors used the PLOS ONE formatting style in the revised manuscript.

Author response: The ethics statement only appears in the Methods section of our manuscript.

3. We note that you have indicated that there are restrictions to data sharing for this study. For studies involving human research participant data or other sensitive data, we encourage authors to share de-identified or anonymized data. However, when data cannot be publicly shared for ethical reasons, we allow authors to make their data sets available upon request. For information on unacceptable data access restrictions, please see http://journals.plos.org/plosone/s/data-availability#loc-unacceptable-data-access-restrictions .

b) If there are no restrictions, please upload the minimal anonymized data set necessary to replicate your study findings to a stable, public repository and provide us with the relevant URLs, DOIs, or accession numbers. Please see http://www.bmj.com/content/340/bmj.c181.long for guidelines on how to de-identify and prepare clinical data for publication. For a list of recommended repositories, please see https://journals.plos.org/plosone/s/recommended-repositories . You also have the option of uploading the data as Supporting Information files, but we would recommend depositing data directly to a data repository if possible.

Author response: According to the ethical review committee, the data contains potentially identifying characteristics as well as sensitive patient information and cannot be shared publicly. All data requests can be sent to the Director of the School of Medicine (Dr. Mulugeta Ayalew at mulugetaayalew336@gmail.com), or the Chair, the Ethical Review Board (prof. Abebe Muche at abemuche14@gmail.com).

Reviewer #1:

Dear Reviewer, we appreciate the valuable comments, suggestions, and requirements you provided, and we have addressed them in our responses as outlined below.

Lines 28-29: How did you selected your study participant?

Author response: Participants were selected using a systematic random sampling method with a skip interval of one. We included this sentence in the revised manuscript (L 88-90).

Lines 32-33: Why not you included OR and CIs?

Author response: The authors modified the revised manuscript by including OR and CIs in the abstract.

Line 34: What is your reference to say high?

Author response: Dear reviewer, thank you. The study found a high prevalence of anxiety among cancer patients, compared to the previous studies, the reason we said "a high prevalence" is to emphasize that the current study observed a greater percentage or rate of anxiety among cancer patients compared to earlier research. It suggests that the anxiety levels among cancer patients in the present study were more significant or more frequent than what was reported in previous studies. This could be due to factors like differences in sample size, methodology, population characteristics, or even changes in the healthcare environment.

Lines 45-62: Your background is too short to really bring or describe the problem under consideration. Please try to discuss your background in chronological order focusing ion Ethiopian context.

Author response: Dear reviewer, thank you for your comment. As per your request, the authors added ideas and discussed the background in chronological order focusing on the Ethiopian context.

Lines 75-77: Why not you considered predictor factors while you calculating your sample size or double population formula?

Author response: Dear reviewer, while we assessed factors associated with anxiety among cancer patients, predictor factors were not considered in the sample size calculation due to the simplicity of the study design. Our primary objective was to focus on estimating the prevalence of anxiety.

Line 81: Cannot be inclusion criteria.

Author response: we removed it from the revised manuscript.

Lines 126-130: For how many days you gave the training? How you did these checkups?

Author response: The quality of data was ensured through the training of data collectors, close supervision, and prompt feedback. The training, which lasted one day, included instruction on interview techniques in accordance with the prepared tool. The checkups were done every other day.

Lines 134-135: Is it bi-variate or bi-variabele and multivariate or multivariable logistic regration? Which selection method did you use?

Author response: Thank you for the question, we used bivariable/multivariable in the analysis, and now we mentioned it the revised manuscript.

Lines 138-139: Have you checked for collinearity? If so what was the result? And what was the result of goodness of fit?

Author response: Dear reviewer, for this study, the Hosmer and Lemeshow goodness-of-fit test was applied, yielding a value of 0.414. Multicollinearity was also assessed using the Variance Inflation Factor (VIF) and tolerance through a pseudo-linear regression analysis. All independent variables had VIF values less than 5 and tolerance values greater than 0.1, indicating the absence of multicollinearity (L143-146).

Lines 148-149: You can put this in the bracket including CI.

Author response: Amendment is done. Please refer to (L169-170)

Line 163: I think these findings are too inflated?

Author response: Thank you for your observation. We understand how these findings might appear elevated at first glance. In our evaluation, the higher figures may be due to

1. The prevalence rates reported were obtained using validated tools like the GAD-7, which is a standard instrument for assessing anxiety in clinical settings. We ensured to follow proper diagnostic criteria and a robust sample size to enhance the accuracy of the findings

2. The higher rates in our study could also be attributed to factors such as the severity of illness in our sample or the stage of follow-up during which the patients were assessed. Cancer follow-up visits can be a particularly stressful time for patients, heightening anxiety levels.

While these numbers may seem elevated, they reflect the real and significant psychological challenges faced by cancer survivors. These insights are critical for improving patient care and guiding future mental health interventions.

Lines 177: What is the difference between RR and OR? How to interpret these two? Please read. Do think your interpretation correct?

Author response: thank you very much for your comment. We rewrite it again please refer it L180-190 in the revised manuscript.

Line 192: What is cotta sampling procedures?

Author response: In the discussion section, we originally used the term 'cotta sampling procedure' as employed by the authors of a referenced article to compare our findings. However, to avoid ambiguity and enhance clarity, we replaced it with 'non-systematic random sampling' in the revised manuscript.

Lines 228-234: For whom you recommending?

Author response: Thank you. We now, in the revised manuscript put specific recommendations to the specific organization (L236-240).

Line 236: What was your reference to say high?

Author response: Dear reviewer, as we tried to elaborate in the question raised at the abstract section, The study found a high prevalence of anxiety among cancer patients, compared to the previous studies, the reason we said "a high prevalence" is to emphasize that the current study observed a greater percentage or rate of anxiety among cancer patients compared to earlier research. It suggests that the anxiety levels among cancer patients in the present study were more significant or more frequent than what was reported in previous studies. This could be due to factors like differences in sample size, methodology, population characteristics, or even changes in the healthcare environment.

Lines 240-243: What is the difference between abbreviations and acronyms? Are these acronyms or Abbreviations?

Author response: thank you very much for your comment. An abbreviation is a shortened version of a word or phrase, not necessarily pronounced as a word whereas an acronym is a specific type of abbreviation that forms a word from the initial letters of a phrase. Based on this definition we used both abbreviations and acronyms (L248).

Lines 245-247: I thought it was a group work. So why I?

Author response: thank you very much for your comment. We appreciate your comment. The authors amended the revised manuscript based on your recommendation.

Additional comments

There are a lots of grammatical and sentence clarity errors. Please improve grammar and spelling errors. Again you have to include statement of declaration, conflict of interest, author contributions and funding statement. Unless it is incomplete manuscript. I will recommend to carefully read authors guideline.

Author response: Dear review, thank you for your comment. The author edited the paper and made clear all the grammar and sentence clarity issues in the revised manuscript. The statement of declaration, conflict of interest, author contributions, and funding statement are revised according to the journal authors' submission guidelines.

Reviewer #2:

Dear Reviewer, we appreciate the valuable comments, suggestions, and requirements you provided, and we have addressed them in our responses as outlined below.

Reviewer #2: Avery good study the study format is very clear, not very wordy I have just comments:

-Comment 1: The abbreviations I prefer to be put at the beginning of the study, not at the end.

Authors’ response: Dear Reviewer, Thank you for your valuable feedback. In accordance with the journal's author guidelines, the abbreviations are placed after the conclusion section. We appreciate your understanding, and we will ensure that all formatting aligns with the journal's requirements.

Comment 2: References: I wish to cite more recent references (2024/2023/2022).

Authors’ response: Dear Reviewer, Thank you for your insightful comment. While we acknowledge the importance of using recent citations, we chose to include certain references because they provide fundamental insights and key findings that remain highly relevant to our study. These sources were carefully selected for their contribution to the field and the strength of their findings, which we believe add significant value to our work.

---

## [Decision Letter · Decision Letter 1]

20 Nov 2024

PONE-D-24-31541R1Anxiety among Cancer Patients: A Cross-Sectional Study at the University of Gondar Comprehensive Specialized HospitalPLOS ONE

Dear Dr. Menberu,,

Thank you for submitting your manuscript to PLOS ONE. After careful consideration, we feel that it has merit but does not fully meet PLOS ONE’s publication criteria as it currently stands. Therefore, we invite you to submit a revised version of the manuscript that addresses the points raised during the review process.

We look forward to receiving your revised manuscript.

Kind regards,

Zelalem Belayneh Muluneh

Academic Editor

PLOS ONE

Journal Requirements:

Reviewers' comments:

Reviewer's Responses to Questions

**Comments to the Author**

1. If the authors have adequately addressed your comments raised in a previous round of review and you feel that this manuscript is now acceptable for publication, you may indicate that here to bypass the “Comments to the Author” section, enter your conflict of interest statement in the “Confidential to Editor” section, and submit your "Accept" recommendation.

Reviewer #1: (No Response)

Reviewer #2: All comments have been addressed

2. Is the manuscript technically sound, and do the data support the conclusions?

Reviewer #1: No

Reviewer #2: Yes

3. Has the statistical analysis been performed appropriately and rigorously? 

Reviewer #1: (No Response)

Reviewer #2: Yes

4. Have the authors made all data underlying the findings in their manuscript fully available?

Reviewer #1: No

Reviewer #2: Yes

5. Is the manuscript presented in an intelligible fashion and written in standard English?

Reviewer #1: No

Reviewer #2: Yes

6. Review Comments to the Author

Reviewer #1: I commented on this paper in a previous review. There are certain issues I recognized while I was checking for whether my first review comments were addressed or not. something like sampling technique, which cannot be corrected after data collection has been changed. For example, the author reported the "QOUTA" sampling technique in a previous submission and now reports it as a "systematic sampling technique. This is unacceptable!

Reviewer #2: Thank you for the prompt answer and adequate reply to all the comments given and wish to continue further research

7. PLOS authors have the option to publish the peer review history of their article (what does this mean? ). If published, this will include your full peer review and any attached files.

**Do you want your identity to be public for this peer review?** For information about this choice, including consent withdrawal, please see our Privacy Policy .

Reviewer #1: **Yes: ** Gemechis Adula

Reviewer #2: **Yes: ** Mohsen M A Abdelhafez

---

## [Author Response · Author response to Decision Letter 2]

27 Nov 2024

Date:27/11/2024

A point-by-point response to editors and reviewers

Title: Anxiety among Cancer Patients: A Cross-Sectional Study at the University of Gondar Comprehensive Specialized Hospital

Manuscript number: PONE-D-24-31541

Subject: Submitting a revised version of the manuscript

We would like to thank the reviewers and editors for sharing their views and novel scholarly experiences. The point-by-point responses for each of the comments, questions, and the revised manuscript are provided in the attached documents.

Thank you for considering our manuscript again

Response to the academic editor

Journal Requirements:

Please review your reference list to ensure that it is complete and correct. If you have cited papers that have been retracted, please include the rationale for doing so in the manuscript text or remove these references and replace them with relevant current references. Any changes to the reference list should be mentioned in the rebuttal letter that accompanies your revised manuscript. If you need to cite a retracted article, indicate the article’s retracted status in the References list and include a citation and full reference for the retraction notice.

Author response: Dear Editor, we reviewed the references, and they are complete and correct. We also declare that we did not cite papers that have been retracted.

Response to Reviewer #1:

Reviewer #1: I commented on this paper in a previous review. There are certain issues I recognized while I was checking for whether my first review comments were addressed or not. something like sampling technique, which cannot be corrected after data collection has been changed. For example, the author reported the "QOUTA" sampling technique in a previous submission and now reports it as a "systematic sampling technique. This is unacceptable!

Author response: Dear reviewer, Thank you for your feedback. We would like to clarify that we did not change our sampling technique. However, we added the following sentence to our manuscript: “Participants were selected using a systematic random sampling method with a skip interval of one” to directly address your previous question: “How did you select your study participants?”

Regarding the term “cotta sampling,” this is unrelated to our sampling technique. Originally, in the discussion section, we used the term ‘cotta sampling procedure’ as it was employed by the authors of a referenced article to compare their findings on the prevalence of anxiety among cancer patients with ours. In response to your question, “What is cotta sampling procedure?”, we replaced it with the term ‘non-systematic random sampling’ in the revised manuscript to ensure clarity.

We hope this explanation addresses your concerns. Please let us know if further clarification is needed.

Reviewer #2:

Reviewer #2: Thank you for the prompt answer and adequate reply to all the comments given and wish to continue further research

Authors’ response: Dear Reviewer, Thank you for your thoughtful comments and feedback. We are pleased to hear that you are satisfied with our responses.

---

## [Decision Letter · Decision Letter 2]

17 Dec 2024

Anxiety among Cancer Patients: A Cross-Sectional Study at the University of Gondar Comprehensive Specialized Hospital

PONE-D-24-31541R2

Dear Dr. Hailu Aragie,

We’re pleased to inform you that your manuscript has been judged scientifically suitable for publication and will be formally accepted for publication once it meets all outstanding technical requirements.

Kind regards,

Zelalem Belayneh Muluneh

Academic Editor

PLOS ONE

Additional Editor Comments (optional):

Reviewers' comments:

Reviewer's Responses to Questions

**Comments to the Author**

Reviewer #2: All comments have been addressed

2. Is the manuscript technically sound, and do the data support the conclusions?

Reviewer #2: Yes

3. Has the statistical analysis been performed appropriately and rigorously? 

Reviewer #2: Yes

4. Have the authors made all data underlying the findings in their manuscript fully available?

Reviewer #2: Yes

5. Is the manuscript presented in an intelligible fashion and written in standard English?

Reviewer #2: Yes

6. Review Comments to the Author

Reviewer #2: Thank you for the production of such a dedicated and informative study which adderessed one of the rarly discussed issued in view of onchology patients which helps to address one ofthe causes of non compliance with the medication.

7. PLOS authors have the option to publish the peer review history of their article (what does this mean? ). If published, this will include your full peer review and any attached files.

**Do you want your identity to be public for this peer review?** For information about this choice, including consent withdrawal, please see our Privacy Policy .

Reviewer #2: **Yes: ** Mohsen M A Abdelhafez

---

## [Editor Report · Acceptance letter]

PONE-D-24-31541R2

PLOS ONE

Dear Dr. Aragie,

I'm pleased to inform you that your manuscript has been deemed suitable for publication in PLOS ONE. Congratulations! Your manuscript is now being handed over to our production team.

Kind regards,

on behalf of

Mr. Zelalem Belayneh Muluneh

Academic Editor

PLOS ONE